# Blinded, randomized trial of sonographer versus AI cardiac function assessment

Bryan He[1], Alan C. Kwan[2], Jae Hyung Cho[2], Neal Yuan[3], Charles Pollick[2], Takahiro Shiota[2], Joseph Ebinger[2], Natalie A. Bello[2], Janet Wei[2], Kiranbir Josan[2], Grant Duffy[2], Melvin Jujjavarapu[4], Robert Siegel[2], Susan Cheng[2,7 ✉], James Y. Zou[1,5,7 ✉] & David Ouyang[2,6,7 ✉]

Artificial intelligence (AI) has been developed for echocardiography[1–3], although it has not yet been tested with blinding and randomization. Here we designed a blinded, randomized non-inferiority clinical trial (ClinicalTrials.gov ID: NCT05140642; no outside funding) of AI versus sonographer initial assessment of left ventricular ejection fraction (LVEF) to evaluate the impact of AI in the interpretation workflow. The primary end point was the change in the LVEF between initial AI or sonographer assessment and final cardiologist assessment, evaluated by the proportion of studies with substantial change (more than 5% change). From 3,769 echocardiographic studies screened, 274 studies were excluded owing to poor image quality. The proportion of studies substantially changed was 16.8% in the AI group and 27.2% in the sonographer group (difference of −10.4%, 95% confidence interval: −13.2% to −7.7%, $P < 0.001$ for non-inferiority, $P < 0.001$ for superiority). The mean absolute difference between final cardiologist assessment and independent previous cardiologist assessment was 6.29% in the AI group and 7.23% in the sonographer group (difference of −0.96%, 95% confidence interval: −1.34% to −0.54%, $P < 0.001$ for superiority). The AI-guided workflow saved time for both sonographers and cardiologists, and cardiologists were not able to distinguish between the initial assessments by AI versus the sonographer (blinding index of 0.088). For patients undergoing echocardiographic quantification of cardiac function, initial assessment of LVEF by AI was non-inferior to assessment by sonographers.

Accurate quantification of cardiac function is necessary for disease diagnosis, risk stratification and assessment of treatment response[4,5]. LVEF, in particular, is routinely used to guide clinical decisions regarding patient appropriateness for a wide range of medical and device therapies as well as interventions including surgeries[4,6,7]. Despite the importance of LVEF assessment in daily clinical practice and clinical research protocols, conventional approaches to measuring LVEF are well recognized as being subject to heterogeneity and variance given that they rely on manual and subjective human tracings[8,9].

Clinical practice guidelines recommend when assessing LVEF based on cardiac imaging—most commonly echocardiography—that the measurements should be performed repeatedly over multiple cardiac cycles to improve precision and account for arrhythmic or haemodynamic sources of variation[10]. Unfortunately, such repeated human measurements are rarely done in practice given logistical constraints present in most clinical imaging laboratories and single tracings or a visual estimation of LVEF is often used as a pragmatic alternative[11,12]. Such an approach is suboptimal for detection of subtle changes in LVEF, which is needed for making important therapeutic decisions (for example, eligibility for continued chemotherapy or defibrillator implantation)[13].

Extending from tremendous progress in the field of AI over the past decade[14], numerous algorithms have been developed with the goal of automating assessment of LVEF in real-world patient care settings[1,3,15]. Although such AI algorithms have demonstrated improved precision on limited retrospective datasets, to date, there are no current cardiovascular AI technologies validated in blinded, randomized clinical trials[16–19]. In addition, human–computer interaction and the effect of AI prompting on clinical interpretations is underexplored in clinical studies. To address this need, we conducted a blinded, randomized non-inferiority clinical trial to prospectively assess the effect of initial assessment by AI versus conventional initial assessment by a sonographer on final cardiologist interpretation of LVEF.

## Cohort characteristics

We enrolled 3,769 transthoracic echocardiogram studies originally performed at an academic medical centre between 1 June 2019 and 8 August 2019; these studies were prospectively re-evaluated by 25 cardiac sonographers (mean of 14.1 years of practice) and 10 cardiologists (mean of 12.7 years of practice). In total, 3,495 studies from 3,035 patients were able to be annotated by sonographers using Simpson's

[1]Department of Computer Science, Stanford University, Palo Alto, CA, USA. [2]Department of Cardiology, Smidt Heart Institute, Cedars-Sinai Medical Center, Los Angeles, CA, USA. [3]Department of Medicine, Division of Cardiology, San Francisco VA, UCSF, San Francisco, CA, USA. [4]Enterprise Information Services, Cedars-Sinai Medical Center, Los Angeles, CA, USA. [5]Department of Biomedical Data Science, Stanford University, Palo Alto, CA, USA. [6]Division of Artificial Intelligence in Medicine, Cedars-Sinai Medical Center, Los Angeles, CA, USA. [7]These authors contributed equally: Susan Cheng, James Y. Zou, David Ouyang. ✉e-mail: susan.cheng@cshs.org; jamesz@stanford.edu; david.ouyang@cshs.org

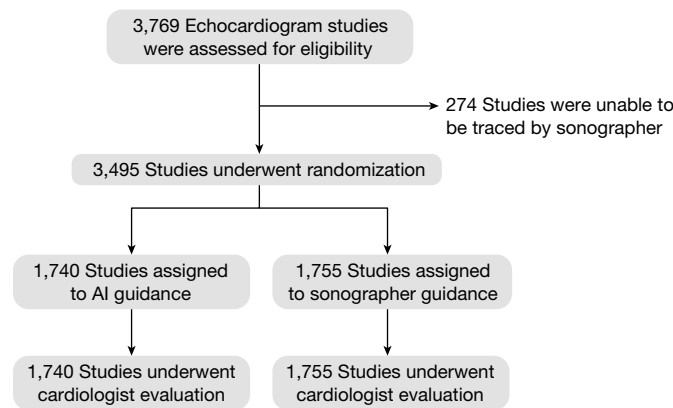

**Fig. 1 | Consort diagram.** Screening, randomization and follow-up.

method of disc calculation of LVEF, and 274 studies were excluded for being of insufficient image quality to contour the left ventricle (Fig. 1). The eligible studies were randomized 1:1 to AI versus sonographer for initial evaluation, with 1,740 studies assigned to the AI group and 1,755 studies assigned to the sonographer group. The baseline characteristics of the two groups were well balanced (Table 1). At a separate workstation and different time, cardiologists were presented with the full echocardiogram study with initial annotations for the final blinded assessment of LVEF.

## Assessment of blinding

After completing each study, cardiologists were asked to predict the agent of initial interpretation. Cardiologists correctly predicted the method of initial assessment for 1,130 (32.3%) studies, incorrectly guessed for 845 (24.2%) studies and were unsure whether initial assessment was provided by AI or sonographers for 1,520 (43.4%) studies. The Bang's blinding index, a metric of blinding in which 0 is perfect blinding and −1 or 1 is perfectly unblinding, was used to assess the trial[20]. A blinding metric between −0.2 and 0.2 is typically considered good blinding, and the blinding index was 0.075 for the sonographer group and 0.088 for the AI group. The bootstrapped confidence interval was consistently within −0.2 and 0.2 ($P > 0.99$).

## Primary outcome

The primary outcome of substantial change between the initial and final assessments occurred in 292 (16.8%) studies in the AI group compared with 478 (27.2%) studies in the sonographer group (difference of −10.4%, 95% confidence interval: −13.2% to −7.7%, $P < 0.001$ for non-inferiority, $P < 0.001$ for superiority) (Table 2). The mean absolute difference between the initial and final assessments of LVEF was 2.79% in the AI group compared with 3.77% in the sonographer group (difference −0.97%, 95% confidence interval: −1.33% to −0.54%, $P < 0.001$ for superiority; Fig. 2).

## Secondary safety outcome

The secondary safety outcome of substantial difference between final cardiologist-adjudicated LVEF compared with the previously clinically reported LVEF occurred in 871 (50.1%) studies in the AI group compared with 957 (54.5%) studies in the sonographer group (difference of −4.5%, 95% confidence interval: −7.8% to −1.2%, $P = 0.008$). The mean absolute difference between previous and final cardiologist assessments was 6.29% in the AI group compared with 7.23% in the sonographer group (difference −0.94%, 95% confidence interval: −1.34% to −0.54%, $P < 0.001$ for superiority; Fig. 2).

**Table 1 | Demographic and imaging study characteristics**

| Characteristic | Total ($n=3,495$) | AI ($n=1,740$) | Sonographer ($n=1,755$) |
|---|---|---|---|
| Age (year) | 66.3±17.0 | 66.1±16.8 | 66.6±17.1 |
| Sex ($n$, %) | | | |
| Male | 1,983 (57%) | 982 (56%) | 1,001 (57%) |
| Female | 1,512 (43%) | 758 (44%) | 754 (43%) |
| Race ($n$, %) | | | |
| Non-Hispanic white | 2,041 (58%) | 1,032 (59%) | 1,009 (57%) |
| Black | 479 (14%) | 230 (13%) | 249 (14%) |
| Hispanic | 405 (12%) | 203 (12%) | 202 (12%) |
| Asian | 273 (8%) | 123 (7%) | 150 (9%) |
| Other | 237 (7%) | 120 (7%) | 117 (7%) |
| Unknown | 38 (1%) | 20 (1%) | 18 (1%) |
| Pacific Islander | 14 (0%) | 8 (0%) | 6 (0%) |
| American Indian | 8 (0%) | 4 (0%) | 4 (0%) |
| Body mass index[a] | 26.5±6.3 | 26.6±6.3 | 26.5±6.2 |
| Comorbidities ($n$, %) | | | |
| Hypertension | 2,019 (58%) | 990 (57%) | 1,029 (59%) |
| Diabetes | 884 (25%) | 441 (25%) | 443 (25%) |
| Coronary artery disease | 1,099 (31%) | 547 (31%) | 552 (31%) |
| Chronic kidney disease | 882 (25%) | 460 (26%) | 422 (24%) |
| Atrial fibrillation | 867 (25%) | 450 (26%) | 417 (24%) |
| Previous stroke | 459 (13%) | 225 (13%) | 234 (13%) |
| Previous clinical EF | 58.1±14.3 | 58.1±14.2 | 58.0±14.4 |
| Method of LVEF evaluation ($n$, %) | | | |
| Single plane (A4C) | 2,249 (64%) | 1,107 (64%) | 1,142 (65%) |
| Biplane | 1,246 (36%) | 633 (36%) | 613 (35%) |
| Study quality ($n$, %) | | | |
| Poor | 648 (19%) | 314 (18%) | 334 (19%) |
| Adequate | 1,725 (49%) | 875 (50%) | 850 (48%) |
| Good | 236 (7%) | 114 (7%) | 122 (7%) |
| Not specified | 886 (25%) | 437 (25%) | 449 (26%) |
| Location ($n$, %) | | | |
| Inpatient | 2,067 (59%) | 1,033 (59%) | 1,034 (59%) |
| Outpatient | 1,428 (41%) | 707 (41%) | 721 (41%) |

A4C, apical-4-chamber; EF, ejection fraction. [a]Body mass index missing in 52 studies.

## Other outcomes and subgroup analyses

The reduction in the primary end point with the AI group was consistent across all major subgroups (Table 3). Between the initial and final assessments, 1,100 (63.2%) studies in the AI group and 1,218 (69.4%) studies in the sonographer group had any change (difference of −6.2%, 95% confidence interval: −9.3% to −3.1%, $P < 0.001$ for superiority). Sonographers took a median of 119 s (interquartile range (IQR): 77–173 s) to assess and annotate LVEF. Cardiologists took a median of 54 s (IQR: 31–95 s) to adjudicate initial assessments in the AI group and a median of 64 s (IQR: 36–108 s) to adjudicate initial assessments in the sonographer group. The mean difference in sonographer time between AI and sonographer groups was −131 s (95% confidence interval: −134 to −127 s, $P < 0.001$). The mean difference in cardiologist time between AI and sonographer groups was −8 s (95% confidence interval: −12 to −4 s, $P < 0.001$).

We additionally assessed the frequency of changes from initial to final assessment crossing a clinically meaningful threshold (that is, LVEF of 35% for consideration of implantable defibrillator therapy) post-hoc. In the AI group, 22 of 1,740 (1.3%) studies crossed the 35%

## Table 2 | Efficacy and safety outcomes

| Outcome | AI (*n*=1,740) | Sonographer (*n*=1,755) | Mean difference (95% confidence interval) | *P* value |
|---|---|---|---|---|
| Primary efficacy outcome: initial versus final assessment | | | | |
| Substantial change | 292 (16.8%) | 478 (27.2%) | −10.5% (−13.2% to −7.7%) | <0.001[a] |
| Mean absolute difference | 2.79±5.53 | 3.77±5.22 | −0.97 (−1.31 to −0.61) | <0.001 |
| Key secondary safety outcome: final versus previous cardiologist assessment | | | | |
| Substantial change | 871 (50.1%) | 957 (54.5%) | −4.5% (−7.8% to −1.2%) | 0.008 |
| Mean absolute difference | 6.29±5.94 | 7.23±6.18 | −0.94 (−1.34 to −0.54) | <0.001 |
| Other secondary outcomes | | | | |
| Sonographer time (s), median (IQR) | 0 (0–0) | 119 (77–173) | −131 (−134 to −127) | <0.001 |
| Cardiologist time (s), median (IQR) | 54 (31–95) | 64 (36–108) | −8 (−12 to −4) | <0.001 |
| Any change | 1,100 (63.2%) | 1,218 (69.4%) | −6.2% (−9.3% to −3.1%) | <0.001 |

[a]For both non-inferiority and superiority tests, all other tests were for superiority. Fisher's exact test was used for categorial outcomes and double-sided Student's *t*-test was used for quantitative outcomes.

threshold between initial and final cardiologist assessments. In the sonographer group, 54 of 1,755 (3.1%) studies crossed the threshold between initial and final assessments (*P* = 0.0004, Chi-squared test).

## Discussion

In this trial of board-certified cardiologists adjudicating clinical transthoracic echocardiographic exams, AI-guided initial evaluation of LVEF was found to be non-inferior and even superior to sonographer-guided initial evaluation. After blinded review of AI versus sonographer-guided

LVEF assessment, cardiologists were less likely to substantially change the LVEF assessment for their final report with initial AI assessment. Furthermore, the AI-guided assessment took less time for cardiologists to overread and was more consistent with cardiologist assessment from the previous clinical report. Although not the first trial of AI technology in clinical cardiology[21–23], to our knowledge, this study represents the first blinded implementation of a randomized trial in this space.

In addition to prospectively evaluating the impact of AI in a clinical trial, our study represents the largest test–retest assessment of clinician variability in assessing LVEF to date. The degree of human variability

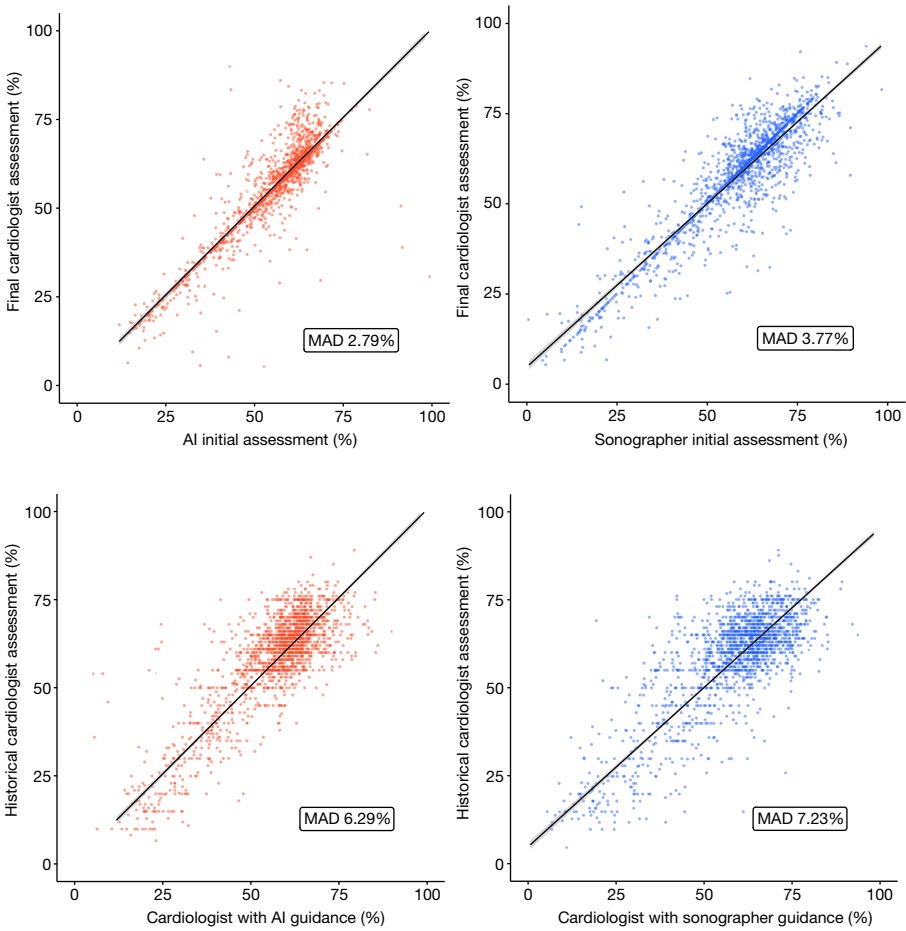

**Fig. 2 | Comparison of AI versus sonographer guidance on cardiologist assessment and difference between final versus previous cardiologist assessments.** Dots represent individual studies and lines represent the lines of best fit. MAD, mean absolute difference.

**Table 3 | Subgroup analysis**

| Subgroup | AI (*n*) | AI (MAD) | Sonographer (*n*) | Sonographer (MAD) | Difference (95% confidence interval) |
|---|---|---|---|---|---|
| Method of LVEF evaluation | | | | | |
| Single plane | 1,107 | 3.20±6.41 | 1,142 | 4.38±5.75 | −1.19 (−1.69 to −0.68) |
| Biplane | 633 | 2.09±3.37 | 613 | 2.61±3.77 | −0.52 (−0.92 to −0.11) |
| Race | | | | | |
| White | 1,032 | 2.58±5.36 | 1,009 | 3.71±5.18 | −1.13 (−1.57 to −0.66) |
| Black | 230 | 3.91±7.59 | 249 | 4.00±5.27 | −0.08 (−1.22 to 1.14) |
| Hispanic | 203 | 2.44±4.26 | 202 | 3.39±4.57 | −0.95 (−1.82 to −0.09) |
| Asian | 123 | 3.11±5.44 | 150 | 4.29±6.04 | −1.18 (−2.55 to −0.23) |
| Other | 152 | 2.77±4.11 | 145 | 3.74±5.26 | −0.97 (−2.07 to −0.08) |
| Sex | | | | | |
| Male | 982 | 2.75±5.92 | 1,001 | 3.67±5.18 | −0.92 (−1.40 to −0.42) |
| Female | 758 | 2.85±4.97 | 754 | 3.89±5.26 | −1.04 (−1.56 to −0.52) |
| Image quality | | | | | |
| Poor | 314 | 4.22±7.12 | 334 | 4.27±5.92 | −0.05 (−1.04 to 0.97) |
| Adequate | 875 | 2.45±5.36 | 850 | 3.53±5.01 | −1.08 (−1.56 to −0.58) |
| Good | 114 | 2.01±3.15 | 122 | 3.51±5.11 | −1.51 (−2.62 to −0.45) |
| Not specified | 437 | 2.66±4.82 | 449 | 3.90±5.02 | −1.24 (−1.89 to −0.58) |
| Location | | | | | |
| Inpatient | 1,033 | 3.09±5.59 | 1,034 | 4.01±5.49 | −0.92 (−1.40 to −0.45) |
| Outpatient | 707 | 2.36±5.41 | 721 | 3.42±4.78 | −1.05 (−1.57 to −0.51) |
| Cardiologist prediction of group | | | | | |
| AI | 557 | 3.64±6.42 | 418 | 3.82±5.09 | −0.18 (−0.91 to 0.54) |
| Sonographer | 427 | 3.38±4.95 | 573 | 4.00±4.62 | −0.62 (−1.21 to 0) |
| Uncertain | 756 | 1.85±4.95 | 764 | 3.56±5.68 | −1.72 (−2.26 to −1.17) |
| Correct prediction | 557 | 3.64±6.42 | 573 | 4.00±4.62 | −0.36 (−0.98 to 0.31) |
| Incorrect prediction | 427 | 3.38±4.95 | 418 | 3.82±5.09 | −0.44 (11.12 to 0.22) |

between repeated LVEF assessments in our study is consistent with previous studies[11,12,24], and the introduction of AI guidance decreased variance between independent clinician assessments. In this trial, we utilized experienced sonographers as an active comparator versus the AI for the initial assessment of LVEF; however, different levels of experience and types of training can change the relative impact of AI compared with clinician judgement. The smaller difference between final and initial assessments, seen in this study for both methods of initial assessment, compared with the difference between final and previous cardiologist assessments highlights the anchoring effect of an initial assessment in practice, and the importance of blinding for quantifying effect size in clinical trials of diagnostic imaging. In both the anchored outcome (comparison of preliminary assessment to final assessment) and independent outcome (comparison of final assessment in the trial versus historical cardiologist assessment), the AI arm showed less variation and more precision in the assessment of LVEF.

Notwithstanding tremendous interest in AI technologies, there have been few prospective trials evaluating their efficacy and effect on clinician assessments. Important clinical trials of AI technology have already shown the efficaciousness of AI in cardiology[21,25]; however, given the difficulty of blinding a diagnostic tool, previous trials are often open-label and compared with a placebo or no diagnostic assistance. Previous works have shown that there can be a Hawthorne effect when studying novel technologies such as AI systems[26,27]. By introducing blinding with an active comparator arm, studies can better distinguish between the effect of the AI technology itself versus the impact of being observed or the act of introducing an intervention. Current FDA-approved technologies for LVEF assessment were not prospectively evaluated with randomization and blinding[16]. By integrating AI into the reporting software, our study sought to minimize bias in assessing the effect size of AI intervention.

To enable effective blinding, we implemented a single cardiac cycle annotation workflow representative of many real-world high-volume echocardiography laboratories. Despite this framework, there was a small signal for cardiologists to be more likely to be correct than incorrect in guessing the agent of initial assessment. However, the blinding index is within the range typically described as good blinding, and regardless of whether the cardiologist thought the initial agent was AI, sonographer or uncertain, the results trended towards improved performance in the AI arm. Our findings of non-inferiority and even superiority of initial AI assessment are encouraging given that AI assessment reduces the time and effort required of the tedious manual processing that is typically required by routine clinical workflows. Given these promising results, further developments of AI could eventually facilitate additional workflows that are required for conducting comprehensive cardiac assessments in routine clinical practice and in accordance with guideline recommendations[28–31].

Several limitations of our trial should be mentioned. First, our study was single centre, reflecting the demographics and clinical practices of a particular population. Nevertheless, the AI model was trained on example images from another centre and the clinical trial was performed as prospective external validation, suggesting generalizability of the AI techniques and workflow. Second, the study was not powered to assess long-term outcomes based on differences in LVEF assessment. Although the results were consistent across subgroups, further analyses are needed to evaluate the impact of video selection, frame selection and intra-provider variability. Third, this trial used previously acquired echocardiogram studies, and although prospectively

# Article

evaluated by sonographers and cardiologists, there can be bias when a different sonographer than the scanning sonographer interprets the images. Last, consistent with findings from most AI studies, we found model performance improvement scales with the number of training examples. Thus, we anticipate that future studies could improve on the AI performance that we observed in the current study by implementing AI models developed based on an even greater number of training examples derived from a broad and diverse cohort of patients. Of note, this clinical trial utilized an AI model entirely trained from an independent site, representing external validation of the model. Effective deployment of AI models in cardiology clinical practice will require additional regulatory oversight, adoption and appropriate use by clinicians, and functional integration with clinical systems, all of which need to be carefully considered and further studied.

In summary, we found that an AI-guided workflow for the initial assessment of cardiac function in echocardiography was non-inferior and even superior to the initial assessment by the sonographer. Cardiologists required less time, substantially changed the initial assessment less frequently and were more consistent with previous clinical assessments by the cardiologist when using an AI-guided workflow. This finding was consistent across subgroups of different demographic and imaging characteristics. In the context of an ongoing need for precision phenotyping, our trial results suggest that AI tools can improve efficacy as well as efficiency in assessing cardiac function. Next steps include studying the effect of AI guidance on cardiac function assessment across multiple centres.

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

## Methods

### Study design and oversight

Cardiologists with board certification in echocardiography were assigned to read independent transthoracic echocardiogram studies randomized to initial assessment by AI versus sonographer. Imaging studies were initially acquired and interpreted clinically by a board-certified cardiologist between 1 June 2019 and 8 August 2019 at Cedars-Sinai Medical Center. Studies were randomly sampled within the time range without regard to patient identity, so that multiple studies from the same patient would be randomized and assessed independently. Sonographers were asked to use their standard clinical practice to annotate the left ventricle for either single-plane or biplane method-of-discs calculation of LVEF[10]. Studies were excluded from randomization if the sonographer was unable to quantify the LVEF owing to inadequate image quality.

Eligible studies were randomly assigned, in a 1:1 ratio, to initial assessment by AI or sonographer and presented to the cardiologists in the standard clinical reporting workflow and software (Siemens Syngo Dynamics VA40D) for adjusting the LV annotation and calculating EF. Although the AI model could annotate every single frame and cardiac cycle, to facilitate blinding, one representative cardiac cycle was annotated and presented to the cardiologist in the AI group. To preserve blinding, the same proportion of single-plane-annotated and biplane-annotated studies was generated for the AI group as the sonographer group.

The trial was designed as a blinded, randomized non-inferiority trial with a prespecified margin of difference by academic study investigators without industry sponsorship or representation in trial design. Approval by the Cedars-Sinai Medical Center Institutional Review Board was obtained before the start of the study. All reading echocardiographers gave informed consent and were excluded from the data analysis. The last author prepared the first draft of the manuscript. The first and last authors had full access to the data. All the authors vouch for the accuracy and completeness of the data and analyses and for the fidelity of the study to the protocol. This study satisfies the requirements set forth by the CONSORT-AI and SPIRIT-AI reporting guidelines[27,31].

### Model design and clinical integration

The architecture and design of the AI model have been previously described[1]. The AI model was trained using 144,184 echocardiogram videos from Stanford Healthcare and was never exposed to videos from the Cedars-Sinai Medical Center. An end-to-end workflow was developed including view classification, LV annotation and LVEF assessment. The AI model was fully embedded within the clinical reporting system at the start of the trial without subsequent changes or adjustments. An in-depth technical description can be found in Supplementary Appendix.

For both sonographers and cardiologists, the entire study (most often between 60 and 120 videos) was shown in the standard clinical reporting software. The study was shown without any annotations to the sonographer, who chose the apical-4-chamber and apical-2-chamber videos and traced the endocardium to assess LVEF. For the cardiologist, the study was shown with one set of annotations (provided by either AI or the sonographer) and can adjust the endocardium to change the reported LVEF (example video 1). Standard method of discs evaluation of the left ventricle, either biplane or single plane depending on sonographer input, was used to calculate LVEF.

### Outcomes assessment

The primary outcome was the change in LVEF between the initial assessment by AI or sonographer and the final cardiologist assessment. The primary outcome was evaluated both as the proportion of studies with substantial change and the mean absolute difference between initial and final assessments. Substantial change was defined as greater than 5% change in LVEF between initial and final assessments. The analysis was performed as randomized and there was no crossover between the two groups.

The duration of time for contouring and adjustment by the sonographer and cardiologist was tracked and compared between the sonographer and AI arms. To assess blinding, cardiologists were asked to predict whether the initial interpretation was by AI, sonographer or unable to tell for each study. A key secondary safety end point was change in final cardiologist-adjudicated LVEF compared with the previous cardiologist-reported LVEF. An additional secondary end point includes the proportion of studies with no change in LVEF between initial and final interpretations.

### Statistical analysis

The trial was designed to test for non-inferiority, with a secondary objective of testing for superiority with respect to the primary end point. Non-inferiority is shown if the lower limit of the 95% confidence interval for the between-group difference in the primary end point was less than 5% (less than the natural variation of test–retest variability in the blinded human assessment of LVEF)[8,24]. With an $\alpha = 0.05$, power of 0.9 and estimating a 5% occurrence rate of substantial change in the AI-driven workflow versus 8% in the sonographer-driven workflow, we estimated a sample size of 2,834 studies needed for statistical power and pre-hoc planned to enrol 3,500 studies for this trial as a buffer for dropout and unforeseen challenges. Non-block randomization was performed once at the beginning of the trial, and the analyses included all imaging studies that underwent randomization (intention-to-treat population). All reported $P$ values for non-inferiority are one-sided, and all reported $P$ values for superiority are two-sided. Bang's blinding index was used to evaluate the quality of blinding during the trial[20]. Statistical analyses were performed with the use of R 4.1.0 (R Foundation for Statistical Computing).

### Reporting summary

Further information on research design is available in the Nature Portfolio Reporting Summary linked to this article.

## Data availability

The AI model was trained using echocardiogram videos from Stanford Healthcare following Stanford IRB protocol 43721 with waiver of individual consent. A set of de-identified Stanford Healthcare echocardiogram videos is publicly available at EchoNet-Dynamic (https://echonet.github.io/dynamic/). The clinical trial was performed at Cedars-Sinai Medical Center under IRB protocol STUDY1707, and the study protocol, statistical analysis plan and de-identified trial results will be available at https://github.com/echonet/blinded_rct.

## Code availability

The code for the AI model is available at GitHub (https://github.com/echonet/dynamic).

**Acknowledgements** No external funding was obtained for this study. We thank the Cedars-Sinai Medical Center Enterprise Information Systems Enterprise Imaging team for their support with clinical integration and deployment.

**Author contributions** B.H., S.C., J.Y.Z. and D.O. designed the clinical trial, study protocol and implementation of the AI model. B.H., G.D. and M.J. engineered technical design and clinical deployment. A.C.K., J.H.C., N.Y., C.P., T.S., J.E., N.A.B., J.W., K.J. and R.S. performed the blinded review of AI versus sonographer assessments. B.H., S.C., J.Y.Z. and D.O. wrote the manuscript with feedback from all authors.

**Competing interests** Stanford University is in the process of applying for a patent application covering video-based deep learning models for assessing cardiac function that lists B.H., J.Y.Z. and D.O. as inventors[1]. All other authors declare no competing interests.

**Additional information**
**Correspondence and requests for materials** should be addressed to Susan Cheng, James Y. Zou or David Ouyang.

# Reporting Summary

## Statistics

For all statistical analyses, confirm that the following items are present in the figure legend, table legend, main text, or Methods section.

| n/a | Confirmed | |
|---|---|---|
| ☐ | ☒ | The exact sample size (*n*) for each experimental group/condition, given as a discrete number and unit of measurement |
| ☐ | ☒ | A statement on whether measurements were taken from distinct samples or whether the same sample was measured repeatedly |
| ☐ | ☒ | The statistical test(s) used AND whether they are one- or two-sided <br> *Only common tests should be described solely by name; describe more complex techniques in the Methods section.* |
| ☐ | ☒ | A description of all covariates tested |
| ☐ | ☒ | A description of any assumptions or corrections, such as tests of normality and adjustment for multiple comparisons |
| ☐ | ☒ | A full description of the statistical parameters including central tendency (e.g. means) or other basic estimates (e.g. regression coefficient) AND variation (e.g. standard deviation) or associated estimates of uncertainty (e.g. confidence intervals) |
| ☐ | ☒ | For null hypothesis testing, the test statistic (e.g. *F*, *t*, *r*) with confidence intervals, effect sizes, degrees of freedom and *P* value noted <br> *Give P values as exact values whenever suitable.* |
| ☒ | ☐ | For Bayesian analysis, information on the choice of priors and Markov chain Monte Carlo settings |
| ☒ | ☐ | For hierarchical and complex designs, identification of the appropriate level for tests and full reporting of outcomes |
| ☒ | ☐ | Estimates of effect sizes (e.g. Cohen's *d*, Pearson's *r*), indicating how they were calculated |

*Our web collection on statistics for biologists contains articles on many of the points above.*

## Software and code

Policy information about availability of computer code

| Data collection | Data was collected from the echocardiography laboratories at Stanford Healthcare and Cedars-Sinai Medical Center. Medical imaging were de-identified from initial DICOM files prior to processing by deep learning algorithm. The deep learning algorithm, written in Python with OpenCV (4.5.1.48), Pytorch (1.8.0) and Torchvision (0.9.0). A full list of dependencies is at https://github.com/echonet/dynamic/blob/master/requirements.txt. Code is available at https://github.com/echonet/dynamic, was used to assess the echocardiogram videos. |
|---|---|
| Data analysis | Statistical analysis is detailed in the statistical analysis plan, to be made available at: https://github.com/echonet/blinded_rct. Analyses were performed in R 4.1.0. |

For manuscripts utilizing custom algorithms or software that are central to the research but not yet described in published literature, software must be made available to editors and reviewers. We strongly encourage code deposition in a community repository (e.g. GitHub). See the Nature Portfolio guidelines for submitting code & software for further information.

## Data

Policy information about [availability of data](availability of data)

All manuscripts must include a [data availability statement](data availability statement). This statement should provide the following information, where applicable:

- Accession codes, unique identifiers, or web links for publicly available datasets
- A description of any restrictions on data availability
- For clinical datasets or third party data, please ensure that the statement adheres to our [policy](policy)

The AI model was trained using echocardiogram videos from Stanford Healthcare following Stanford IRB protocol 43721 with waiver of individual consent. A set of de-identified Stanford Healthcare echocardiogram videos is publicly available at https://echonet.github.io/dynamic/. The clinical trial was performed at Cedars-Sinai Medical Center under IRB protocol STUDY1707, and the study protocol, statistical analysis plan, and de-identified trial results will be available at https://github.com/echonet/blinded_rct. Data requests for identifiable data will be evaluated by the authors to maintain compliance with IRB protocol and data privacy protections. Please email david.ouyang@cshs.org for requests for identifiable data.

## Human research participants

Policy information about [studies involving human research participants and Sex and Gender in Research.](studies involving human research participants and Sex and Gender in Research.)

| | |
|---|---|
| Reporting on sex and gender | Stratified data analysis was reported by sex and the cohort demographics were described in Table 1 |
| Population characteristics | Detailed cohort characteristics given in Table 1 and Supplementary Table 1 |
| Recruitment | Consecutive echocardiograms from August 2019 were to train the model. Sonographers and cardiologists from the Cedars Sinai Medical Center echo lab were recruited for the study and performed evaluation between February 1, 2022 and July 5, 2022. |
| Ethics oversight | The AI model was trained using echocardiogram videos from Stanford Healthcare following Stanford IRB protocol 43721 with waiver of individual consent. The clinical trial was performed at Cedars-Sinai Medical Center under IRB protocol STUDY1707 |

Note that full information on the approval of the study protocol must also be provided in the manuscript.

# Field-specific reporting

Please select the one below that is the best fit for your research. If you are not sure, read the appropriate sections before making your selection.

☒ Life sciences ☐ Behavioural & social sciences ☐ Ecological, evolutionary & environmental sciences

For a reference copy of the document with all sections, see [nature.com/documents/nr-reporting-summary-flat.pdf](nature.com/documents/nr-reporting-summary-flat.pdf)

# Life sciences study design

All studies must disclose on these points even when the disclosure is negative.

| | |
|---|---|
| Sample size | Non-inferiority Design: 8% vs. 5%, alpha of 0.05 and power of 0.9<br>2834 studies needed, pre-planned to enroll 3500 studies as buffer against dropout |
| Data exclusions | A pre-specified run in period was performed where sonographers traced all echocardiogram studies and studies that sonographers could not trace were excluded from randomization. All studies randomized were revaluated by cardiologists. |
| Replication | 95% Confidence interval of reported metrics were determined by boot strapping. Trial was only performed once without replication. |
| Randomization | Studies were individually randomized 1:1 to AI vs. sonographer as agent of initial interpretation. |
| Blinding | Cardiologists were blinded to agent of initial interpretation and asked to guess agent of initial interpretation (AI vs. sonographer) in blinded fashion. |

# Reporting for specific materials, systems and methods

We require information from authors about some types of materials, experimental systems and methods used in many studies. Here, indicate whether each material, system or method listed is relevant to your study. If you are not sure if a list item applies to your research, read the appropriate section before selecting a response.

## Materials & experimental systems

| n/a | Involved in the study |
|---|---|
| ☒ | ☐ Antibodies |
| ☒ | ☐ Eukaryotic cell lines |
| ☒ | ☐ Palaeontology and archaeology |
| ☒ | ☐ Animals and other organisms |
| ☐ | ☒ Clinical data |
| ☒ | ☐ Dual use research of concern |

## Methods

| n/a | Involved in the study |
|---|---|
| ☒ | ☐ ChIP-seq |
| ☒ | ☐ Flow cytometry |
| ☒ | ☐ MRI-based neuroimaging |

## Clinical data

Policy information about clinical studies
All manuscripts should comply with the ICMJE guidelines for publication of clinical research and a completed CONSORT checklist must be included with all submissions.

| | |
|---|---|
| Clinical trial registration | https://clinicaltrials.gov/ct2/show/NCT05140642 |
| Study protocol | Study protocol in supplementary materials |
| Data collection | Data collected from Cedars Sinai Medical Center between February 2, 2021 and July 5, 2022 |
| Outcomes | Primary Outcome: Frequency and degree of change from initial (AI vs. sonographer) assessment to final cardiologist assessment<br>Substantial change defined as more than 5% LVEF<br><br>Secondary Outcomes:<br>Cardiologist Prediction of Agent of Initial Assessment (Blinding Assessment)<br>Sonographer Time<br>Cardiologist Time<br>Change from Historical Cardiologist Assessment |

