## [Peer Review File · Nature]

Manuscript Title: Blinded Randomized Trial of Sonographer vs AI Cardiac Function Assessment

Reviewer Comments & Author Rebuttals

Reviewer Reports on the Initial Version:

Referees' comments:

Referee #1 (Remarks to the Author):

This is a retrospective, single-center, randomized study that compared AI vs sonographer determined ejection fraction and how that impacted the cardiologist's final reporting of this metric. The AI led to less change of the cardiologist's assessment and saved ~2 minutes compared with the sonographer's determination. This non-inferiority and superiority of the primary endpoint essentially validates the AI tool for initial screening of ejection fraction, which has been done via other means without this trial construct, such as the NPJ Digital Medicine 2020 report on "Deep learning interpretation of echocardiograms" by Zhou and colleagues, a co-author of this paper. There are some grand claims that need to be toned down. This is certainly not the first randomized trial of AI technology applied to clinical cardiology. Beyond the fact that it is retrospective, there are prospective randomized trials such as the Mayo Clinic ECG (Yao et al, 2001) and others in acute heart failure, blood pressure management (reviewed in Plana, JAMA Network Open, Sept 2022). Moreover, a retrospective in silico study is hardly a randomized, real-world assessment of AI. The authors emphasize the "blinding" aspect, but that would be far more important in a prospective trial. Many echo labs do not have the sonographer compute ejection fraction and that metric is solely read out by the cardiologist. This practical point is not commented upon in the paper. For that common practice, a direct comparison of AI with cardiologist assessment of EF would be more meaningful.

Referee #2 (Remarks to the Author):

The paper clearly presents the results of a clinical trial to assess the non-inferiority of an AI-based method for initial assessment of left ventricular ejection fraction, one step to facilitate the work of cardiologists in interpreting an echocardiography. The AI method was previously developed and published.

The work is novel and an important step towards validating existing technology for their use in the clinical space.

The size of the experiment is appropriate, and the quality of the presentation is good. There is a concern regarding the term prospectively re-evaluated used in the Results. I would define this study, from the title, as a retrospective study as no new data is collected with the study. Once this is clearly stated, the term "prospectively re-evaluated" will be clear and correct in that context.

No major concern on the standard statistical methods applied.

The impact of this paper should be clarified. While this is an important step towards validating this AI technique, the actual AI engine has already been presented. This study is not the final step, since it does not demonstrate the utility of this AI technique in a real clinical setting prospectively. Thus, the impact of this publication may be limited.

Another concern, to be better discussed, regards assessment of blindness: despite the cardiologist were not able to always determine if the initial assessment was done by AI or a sonographer, they were at least more likely to be correct than wrong (not perfectly blind to it). Not a big concern, but this should be better discussed in Discussion.

Similarly, the anchor effect seems to be strong from the presented results, and given the non-perfect blindness, this may be a concern in interpreting the results. At least a deeper discussion on this point would be needed.

Minor comments:

- in Abstract: "The mean absolute difference from between final and prior...": sentence to be re-phrased and clarified, the "prior cardiology assessment" was clear to the reader only after reading the Methods. It should be clarified what we mean to a non-expert when this results are presented in the Abstract.

- "as be subject to heterogeneity" -> "as being subject to"?

Referee #3 (Remarks to the Author):

The paper describes a prospective, blinded randomized controlled trial to evaluate a previously published AI algorithm for assessing left ventricular ejection fraction (LVEF), by comparing it against sonographer initial assessment. The study is well designed and well carried out. The statistical analyses are appropriate and results are convincing. The paper is very well written, with clearly laid out motivations, clearly explained approach and conclusions, as well as thoughtful descriptions of study limitations.

Even though the trial being single center is a major limitation, and furthermore it was limited to one particular AI model, the fact that the model was trained on data from a completely different population does land value to the results. As the authors recognized, more trials should be conducted in the near future, on more diverse trial population, and to evaluate AI models trained on broader set of data. Nonetheless, as the first study of its kind, the findings presented in this paper represent a significant initial step forward and are of great importance to the field.

Author Rebuttals to Initial Comments:

We appreciate the valuable feedback from the Editor and Reviewers. We detail below the changes to our manuscript that we have made in response to the helpful comments and suggestions provided. We believe manuscript is substantially improved as a result.

Comments from the Reviewer 1

This is a retrospective, single-center, randomized study that compared AI vs sonographer determined ejection fraction and how that impacted the cardiologist's final reporting of this metric. The AI led to less change of the cardiologist's assessment and saved ~2 minutes compared with the sonographer's determination. This non-inferiority and superiority of the primary endpoint essentially validates the AI tool for initial screening of ejection fraction, which has been done via other means without this trial construct, such as the NPJ Digital Medicine 2020 report on "Deep learning interpretation of echocardiograms" by Zhou and colleagues, a co-author of this paper.

Reply: We appreciate the thoughtful comments provided by the Reviewer. We would like to clarify that this was a randomized, blinded trial that involved 25 sonographers and 10 cardiologists with pre-specified endpoints (available online prior to trial initiation: <https://clinicaltrials.gov/ct2/show/NCT05140642>). Informed consent was obtained from all clinicians, similar to the Mayo ECG clinical trial, and all adjudications were done for the clinical trial.

From Yao et al. Nature Medicine, 2021 (<https://www.nature.com/articles/s41591-021-01335-4>):
A total of 358 of 525 eligible primary care clinicians provided informed consent. No clinicians withdrew consent during the study period. Patients themselves were not considered 'subjects' in this study (as such, patients themselves were not consented for participation).

From our paper:

The trial was designed as a blinded, randomized non-inferiority trial with prespecified margin of difference by academic study investigators without industry sponsorship or representation in trial design. Approval by the Cedars-Sinai Medical Center Institutional Review Board was obtained prior to the start of the study. All reading echocardiographers gave informed consent.

Specifically, this trial was presented as a late breaking clinical trial at the European Society of Cardiology 2022 meeting (<https://www.escardio.org/The-ESC/Press-Office/Press-releases/Artificial-intelligence-assessment-of-heart-function-is-superior-to-sonographer-assessment>), highlighting its reception as a blinded, randomized clinical trial.

An important aspect the reviewer brings up is that the patient videos were collected originally from 2019 while the sonographer re-evaluation and cardiologist evaluation was done in 2022.

From the results: 3769 transthoracic echocardiogram studies originally performed at an academic medical center between June 1, 2019 and August 8, 2019 were prospectively re-evaluated by twenty-five cardiac sonographers (mean of 14.1 years of practice) and ten cardiologists (mean of 12.7 years of practice).

Trial Design

ESC Congress 2022 Barcelona
Onsite & Online

This is an important feature of our study and we hope to clarify that this was mandated by our IRB as the review board thought AI technology was investigational and did not allow the study team to incorporate contemporaneous echocardiograms. We were asked to use previously collected echocardiograms to have the safety endpoint (the same images are evaluated by cardiologists twice and for the sonographer arm, be the largest test-retest of human clinician performance) in anticipation of future clinical deployment. However, even then, both sonographer and cardiologist assessment were done prospectively and within prespecified criteria trial set-up. We think there should not be a significant change in the results if the scanning was contemporaneous as the assessments in clinical practice are independent and stepwise (the way it was done in the trial) and in this trial, assessment of LVEF after scanning by both sonographer and cardiologist was done prospectively and in pre-specified fashion.

We recognize the confusion with descriptors of prospective and retrospective, and to help clarify the manuscript, removed references to prospective and only focus on the trial protocol.

In the abstract: *Artificial intelligence (AI) models have been developed for quantifying cardiac function in echocardiography, although not yet **prospectively** tested with blinding and randomization. To evaluate impact of AI algorithm in the echocardiographic interpretation workflow, we designed a **prospective** blinded, randomized non-inferiority clinical trial of AI vs. sonographer initial assessment of left ventricular ejection fraction (LVEF) (ClinicalTrials.gov number NCT05140642, no outside funding).*

In the discussion: *Several limitations of our trial should be mentioned. First, our study was single center, reflecting the demographics and clinical practices of a particular population. [...] Second, the study was not powered to assess long term outcomes*

based on differences in LVEF assessment. [...]. Third, this trial used previously acquired echocardiogram studies, and while prospectively evaluated by sonographers and cardiologists, there can be bias when a different sonographer than the scanning sonographer interprets the images.

In addition to the particular question and technology studied, we think there is interest in the trial design in regards to randomization and blinding. In particular, it is difficult to blind AI technologies and all prior trials of AI do not use an active comparator (only comparing with nothing). For our trial, to blind and randomize with sonographers (the active comparator), we had to build substantial infrastructure to imbed the AI system within the clinical reporting system. This is in contrast with our prior work with purely in silico validation published in Nature 2020 and npj Digital Medicine, which were not blinded or randomized. Even in other prospective AI trials, there is not a similar ability to blind and as such, randomization is often cluster-randomized by site, rather by individual study.

We clarify that we are not claiming to be the first randomized trial of AI technology in cardiology, but the first blinded randomized trial of AI technology in cardiology. This is worthy of interest as blinding minimizes participant bias and allows for the introduction of an active comparator. EchoNet-RCT is the first trial where the AI technology is compared head to head with a clinician, while prior trials compared with the absence of interpretation. We clarify and reframe the discussion to describe this point:

In the discussion: To our knowledge, this study represents the first blinded and randomized trial of AI technology applied to clinical cardiology.

Has been changed to: *While not the first trial of AI technology in clinical cardiology²² [citation of the Yao et al and Noseworthy et al clinical trials], to our knowledge, this study represents the first **blinded implementation of a randomized trial in this space.***

There are some grand claims that need to be toned down. This is certainly not the first randomized trial of AI technology applied to clinical cardiology. Beyond the fact that it is retrospective, there are prospective randomized trials such as the Mayo Clinic ECG (Yao et al, 2001) and others in acute heart failure, blood pressure management (reviewed in Plana, JAMA Network Open, Sept 2022). Moreover, a retrospective in silico study is hardly a randomized, real-world assessment of AI. The authors emphasize the “blinding” aspect, but that would be far more important in a prospective trial.

Reply: Thank you for the important comments provided by the Reviewer and the opportunity to revise our paper to reflect important precedence and context for our trial. In this revision, we clarify that we are not trying to claim to be the first randomized trial of AI technology in cardiology, but the first blinded randomized trial of AI technology in cardiology. As the reviewer mention, blinding is an important aspect of clinical trials (one that is particularly hard to do for diagnostic tools as it is hard to find a fair active comparator). In both studies described by the author, blood pressure management and ECG screening of low EF, the interventions were open label and unblinded – which can introduce bias as study participants might have either favorable or unfavorable impressions of the intervention.

With the introduction of blinding, trials able to introduce an active comparator (for this trial, the comparison with human sonographer annotation). Without an active comparator, it is difficult to assess whether the effect of the intervention is due to 1) AI technology, 2) the active of being observed (Hawthorne effect), or 3) the result of more intensive follow-up (for example: lead time bias). For example, our prior work showed that simply interacting with a system known to be an AI system changes the behavior of clinicians (<https://arxiv.org/abs/2107.07015>). By introducing blinding, the relative effect of being in a trial vs. the actual AI technology can more teased out.

In the discussion: *While not the first trial of AI technology in clinical cardiology²⁰⁻²² [citation of the Yao et al and Noseworthy et al clinical trials], to our knowledge, this study represents the first blinded implementation of a randomized trial in this space.*

Additionally, in the discussion we add: *Notwithstanding tremendous interest in AI technologies, there have been few prospective trials evaluating their efficacy and impact on clinician assessments. Important clinical trials of AI technology have already shown the efficaciousness of AI in cardiology²²⁻²³, however given the difficulty of blinding a diagnostic tool, previous trials are often open label and compared with a placebo or lack of diagnostic assistance. Prior work have shown that there is can be Hawthorne effect when studying novel technologies like AI systems^{25,26}, by introducing blinding with an active comparator arm, studies can better distinguish between the effect of the AI technology itself vs. the impact of being observed or the act of introducing an intervention. Current FDA cleared technologies for LVEF assessment were not prospectively evaluated with randomization and blinding.²¹⁻²³ By integrating the AI into the reporting software, our study sought to minimize bias in assessing the effect size of AI intervention.*

This was a randomized, blinded trial that prospective involved 25 sonographers and 10 cardiologists with pre-specified endpoints and informed consent was obtained from all clinicians. In particular, we think the design is interesting with the use of an active comparator (the sonographer) which is often not done in prospective trials of AI. With the use of an active comparator, we are able to do study-level randomization (in contrast to site-level cluster randomization which needs to be done when there is no active comparator).

Many echo labs do not have the sonographer compute ejection fraction and that metric is solely read out by the cardiologist. This practical point is not commented upon in the paper. For that common practice, a direct comparison of AI with cardiologist assessment of EF would be more meaningful.

Reply: We appreciate the thoughtful comments provided by the Reviewer. Across the world, there are different practice set-ups, and in Europe and Asia, often the cardiologist is alone in evaluating ejection fraction without an aid. We particularly chose the design of the American model (where sonographers initially interpret and cardiologists finalize) because it facilitated blinding and randomization since there are two independent points of expert clinician contact and allow for the comparison with an active comparator. Prior trials in AI only compare with the lack of assistance or “standard of care”, but the introduction of an active comparator allows more insight into the impact of the AI technology vs. open-label trials.

We sought a strong benchmark of comparison, such that we used a group of highly experienced sonographers as a high bar to compare with. We would imagine less experienced sonographers or a stronger cardiologist comparator could change the difference between AI and non-AI assistance. In this study, the sonographers had an average of 14 years of experience, in which many clinicians would say they become quite good in assessing LVEF; however, we should recognize the heterogeneity and variation that can occur based on provider.

In the discussion: *In addition to prospective evaluating the impact of AI in a clinical trial, our study represents the largest to date test-retest assessment of clinician variability in assessing LVEF. The degree of human variability between repeated LVEF assessments in our study is consistent with prior studies,^{8,9,21} and the introduction of AI guidance decreased variance between independent clinician assessments. In this trial, we utilized experienced sonographers as an active comparator vs. the AI for the initial assessment of LVEF, different levels of experience and types of training can change the relative impact of AI compared to clinician judgement.*

To evaluate heterogeneity among providers, in our supplemental results, we describe how much variation there is for individual clinicians, lighting the reviewers important point in choice of clinician and evaluation.

Supplemental Figure 3: Performance of each individual sonographer vs. AI initial assessment compared to historical assessment. Boxplot of interquartile range (IQR) and median. Whiskers truncated beyond 20% difference.

Comments from the Reviewer 2

The paper clearly present the results of a clinical trial to assess the non-inferiority of an AI-based method for initial assessment of left ventricular ejection fraction, one step to facilitate the work of cardiologists in interpreting an echocardiography. The AI method was previously developed and published. The work is novel and an important step towards validating existing technology for their use in the clinical space.

Reply: Thank you for your review of our trial and we appreciate your feedback on how to improve the presentation of the results. We think there is particular value in evaluating AI technologies in blinded and randomized fashion.

The size of the experiment is appropriate, and the quality of the presentation is good. There is a concern regarding the term prospectively re-evaluated used in the Results. I would define this study, from the title, as a retrospective study as no new data is collected with the study. Once this is clearly stated, the term “prospectively re-evaluated” will be clear and correct in that context.

Reply: Thank you for your review of our trial and we appreciate your feedback. An important aspect the reviewer brings up is that the patient videos were collected originally from 2019 while the sonographer re-evaluation and cardiologist evaluation was done in 2022.

From the results: *3769 transthoracic echocardiogram studies originally performed at an academic medical center between June 1, 2019 and August 8, 2019 were prospectively*

re-evaluated by twenty-five cardiac sonographers (mean of 14.1 years of practice) and ten cardiologists (mean of 12.7 years of practice).

This is an important feature of our study – in fact, mandated by our IRB as the review board thought AI technology was investigational and did not allow the study team to incorporate contemporaneous echocardiograms – however both sonographer and cardiologist assessment were done prospectively and within prespecified criteria trial set-up. Notably, this allows us to have the safety endpoint (since the same images are evaluated by cardiologists twice and for the sonographer arm, be the largest test-retest of human clinician performance).

We recognize the confusion with descriptors of prospective and retrospective, and to help clarify the manuscript, removed references to prospective and only focus on the trial protocol.

*In the abstract: Artificial intelligence (AI) models have been developed for quantifying cardiac function in echocardiography, although not yet **prospectively** tested with blinding and randomization. To evaluate impact of AI algorithm in the echocardiographic interpretation workflow, we designed a **prospective** blinded, randomized non-inferiority clinical trial of AI vs. sonographer initial assessment of left ventricular ejection fraction (LVEF) (ClinicalTrials.gov number NCT05140642, no outside funding).*

*In the discussion: Several limitations of our trial should be mentioned. First, our study was single center, reflecting the demographics and clinical practices of a particular population. [...] Second, the study was not powered to assess long term outcomes based on differences in LVEF assessment. [...]. **Third, this trial used previously acquired echocardiogram studies, and while prospectively evaluated by sonographers and cardiologists, there can be bias when a different sonographer than the scanning sonographer interprets the images.***

We also change to the title to highlight the use of clinical echocardiograms reflecting the difference between the prospective assessment and the historically obtained imaging.

Blinded, Randomized Controlled Trial of Sonographer vs. Artificial Intelligence Assessment of Cardiac Function in Clinical Acquired Echocardiograms

No major concern on the standard statistical methods applied.

Reply: Thank you for your feedback

The impact of this paper should be clarified. While this is an important step towards validating this AI technique, the actual AI engine has already been presented. This study is not the final step, since it does not demonstrate the utility of this AI technique in a real clinical setting prospectively. Thus, the impact of this publication may be limited.

Reply: Thank you for your feedback and we definitely agree with the challenges presented given the evolving landscape of evaluating AI technologies. For our trial, we had initially proposed an entirely prospective evaluation, however was recommended revision by the IRB because of the concern of deployment of an entirely new technology and asked us to consider the current design which allows us to evaluate the safety of the technology (as well as assess human clinician test-retest variation). We see this as a necessary intermediate step, as many medical centers are not yet comfortable with the idea of AI technology, even with clinician oversight.

For our trial, to blind and randomize with sonographers (the active comparator), we had to build substantial infrastructure to imbed the AI system within the clinical reporting system. This is something we hope is worthy of interest to the general AI and cardiology audience and an additional necessary step for final deployment. As an aside, we note that after presenting the integration at ESC, the clinical software system vendors approached the study team to ask about integration of the AI technology because this is an area of active interest and necessary consideration prior to deployment. Given the combination of regulatory as well as technical deployment challenges, we hope our study is a step towards ultimate deployment of AI technology in cardiology, and we seek to clarify this in the revised discussion:

In the discussion: Several limitations of our trial should be mentioned. First, our study was single center, reflecting the demographics and clinical practices of a particular population. [...] Third, this trial used previously acquired echocardiogram studies, and while prospectively evaluated by sonographers and cardiologists, there can be bias when a different sonographer than the scanning sonographer interprets the images. Finally, consistent with findings from most AI studies, we found model performance improvement scales with the number of training examples. Thus, we anticipate that future studies could improve upon the AI performance that we observed in the current study by implementing AI models developed based on an even greater number of training examples derived from a broad and diverse cohort of patients. Notably, this clinical trial utilized an AI model entirely trained from an independent site, representing external validation of the model. Final deployment of AI models in cardiology will require additional regulatory oversight, clinician buy-in, and deep integration with clinical systems that need to be further studied.

Another concern, to be better discussed, regards assessment of blindness: despite the cardiologist were not able to always determine if the initial assessment was done by AI or a sonographer, they were at least more likely to be correct than wrong (not perfectly blind to it). Not a big concern, but this should be better discussed in Discussion.

Reply: Thank you for feedback and the opportunity to improve the presentation of the results. Blinding is an important aspect of randomized trial, and in particular we took particular emphasis to try to maximize blinding (by showing the same proportion of similar types of annotation and display them in the clinical software system to minimize variation from sonographer annotations). Additionally, we note that there are no prior blinded clinical trials of AI technology and many prospective clinical trials of even therapeutics do not formally assess blinding (e.g., in

vaccine trials, the presence of side effects of sore arm and etc likely create a small degree of unblinding that could bias the participants).

Our blinding index was between -0.2 and 0.2, which is typically considered good blinding (Bang et al. Control Clin Trials, 2004). and within the range of statistical noise if it was randomly assessed. For example, even in the setting of guessing heads or tails, one will not always be incorrect, and there is a statistically acceptable range of correct guesses that can happen from random sampling and due to noise alone.

We believe there is particular value in evaluating AI technologies in blinded and randomized fashion, and try to discuss this in more detail in the revised discussion:

In the discussion: By integrating the AI into the reporting software, our study sought to minimize bias in assessing the effect size of AI intervention. To enable effective blinding, we implemented a single cardiac cycle annotation workflow representative of many real-world high-volume echocardiography laboratories. Despite this framework, there was a small signal for cardiologists to be more likely to be correct than incorrect in guessing the agent of initial assessment. However, the blinding index is within the range typically described as good blinding, and regardless whether the cardiologist thought the initial agent as AI, sonographer, or uncertain, the results trended towards improved performance in the AI arm.

Additionally, in new analyses, we show that irrespective of whether the cardiologists were able correctly or incorrectly guess the initial agent of interpretation, the trend was towards improved performance by AI. These subset analyses are new/ad hoc, so not powered for significance, but trend in the same direction and have minimal heterogeneity.

From Table 3:

Subgroup	AI n	AI MAD	Sonographer n	Sonographer MAD	Difference (95% CI)
Cardiologist Prediction of Group					
AI	557	3.64±6.42	418	3.82±5.09	-0.18 (-0.91 to 0.54)
Sonographer	427	3.38±4.95	573	4.00±4.62	-0.62 (-1.21 to 0.00)
Uncertain	756	1.85±4.95	764	3.56±5.68	-1.72 (-2.26 to -1.17)
Correct Pred	557	3.64±6.42	573	4.00±4.62	-0.36 (-0.98 to 0.31)
Incorrect Pred	427	3.38±4.95	418	3.82±5.09	-0.44 (-1.12 to 0.22)

Similarly, the anchor effect seems to be strong from the presented results, and given the non-perfect blindness, this may be a concern in interpreting the results. At least a deeper discussion on this point would be needed.

Reply: Thank you for your review of our trial and we appreciate the feedback. We recognize that there is a small amount of signal that could result imperfect blinding but within the range of possible statistical noise. Even in this setting, we note that there was no significant difference between subgroups by whether cardiologist through the performance was by AI, sonographer or uncertain, and all groups trended to improved performance in the AI arm (although not statistically powered in the subgroup analysis), as well as whether the cardiologist was incorrect or correct in guessing the initial annotator.

From Table 3:

Subgroup	AI n	AI MAD	Sonographer n	Sonographer MAD	Difference (95% CI)
Cardiologist Prediction of Group					
AI	557	3.64±6.42	418	3.82±5.09	-0.18 (-0.91 to 0.54)
Sonographer	427	3.38±4.95	573	4.00±4.62	-0.62 (-1.21 to 0.00)
Uncertain	756	1.85±4.95	764	3.56±5.68	-1.72 (-2.26 to -1.17)
Correct Pred	557	3.64±6.42	573	4.00±4.62	-0.36 (-0.98 to 0.31)
Incorrect Pred	427	3.38±4.95	418	3.82±5.09	-0.44 (-1.12 to 0.22)

Additionally, the results trended in the same direction with anchoring (the primary result gave cardiologist the initial interpretation) as without anchoring (the key safety endpoint comparison was comparing with standard clinical measurement which would not have an anchoring of AI assistance). Human clinician variation, and in particular anchoring is an important issue, often understudied in clinical research, to the degree that there are even limited studies of variance in unanchored human performance. Our study (by comparing just the cardiologist assessment in the sonographer arm with the historical cardiologist assessment) can potentially be useful to the cardiology literature as the largest clinician test-retest evaluation of LVEF. We seek to more broadly discuss these important points in the discussion and are open to feedback:

In the discussion: *In this trial, we utilized experienced sonographers as an active comparator vs. the AI for the initial assessment of LVEF, different levels of experience and types of training can change the relative impact of AI compared to clinician judgement. The smaller difference between final and initial assessment, seen in this study for both methods of initial assessment, compared to the difference between final and prior cardiologist assessment highlights the anchoring effect of an initial assessment in practice - and the importance of blinding for quantifying effect size in clinical trials of diagnostic imaging. In both the anchored outcome (comparison of preliminary to final*

assessment) and independent outcome (comparison of final assessment in the trial vs. historical cardiologist assessment), the AI arm showed less variation and more precision in the assessment of LVEF.

Additionally, we further discuss inter-clinician variability and provide discussion that we observed no differences in model performance by image quality, inpatient vs. outpatient status, or single plane vs. biplane assessment. To clarify these findings, we include additional analyses demonstrating that the AI algorithm's median absolute difference from historical LVEF is smaller than the median absolute difference for 22 of the 27 individual sonographers who provided measurements for the study. In this setting, we believe historical LVEF is the best comparison without anchoring.

Supplemental Figure 3: Performance of each individual sonographer vs. AI initial assessment compared to historical assessment. Boxplot of interquartile range (IQR) and median. Whiskers truncated beyond 20% difference.

Minor comments:

- in Abstract: "The mean absolute difference from between final and prior...": sentence to be re-phrased and clarified, the "prior cardiology assessment" was clear to the reader only after reading the Methods. It should be clarified what we mean to a non-expert when this results are presented in the Abstract.

Reply: Thank you for feedback and we hope this is clarified by changing to:

In the abstract: *The mean absolute difference between **final cardiologist assessment** and **independent prior cardiologist** assessment was 6.29% in the AI group and 7.23% in the sonographer group (difference -0.96%, 95% CI -1.34% to -0.54%, $P < 0.001$ for superiority).*

- "as be subject to heterogeneity" -> "as being subject to"?

Reply: Thank you for feedback and that has been changed.

In the discussion: *Despite the importance of LVEF assessment in daily clinical practice and clinical research protocols, conventional approaches to measuring LVEF are well recognized as **being** subject to heterogeneity and variance given that they rely on manual and subjective human tracings.^{5,6}*

Comments from the Reviewer 3

The paper describes a prospective, blinded randomized controlled trial to evaluate a previously published AI algorithm for assessing left ventricular ejection fraction (LVEF), by comparing it against sonographer initial assessment. The study is well designed and well carried out. The statistical analyses are appropriate and results are convincing. The paper is very well written, with clearly laid out motivations, clearly explained approach and conclusions, as well as thoughtful descriptions of study limitations.

Reply: Thank you for your review of our trial and we appreciate your feedback. We think there is particular value in evaluating AI technologies in blinded and randomized fashion. We recognize that this is an evolving landscape and seek to integrate the feedback on future directions and additional experiments remaining to be done in the field.

Even though the trial being single center is a major limitation, and furthermore it was limited to one particular AI model, the fact that the model was trained on data from a completely different population does land value to the results. As the authors recognized, more trials should be conducted in the near future, on more diverse trial population, and to evaluate AI models trained on broader set of data. Nonetheless, as the first study of its kind, the findings presented in this paper represent a significant initial step forward and are of great importance to the field.

Reply: Thank you for your review of our trial and we appreciate your feedback on how to improve the presentation of the results. We seek to present the trial fairly and integrate discussion of some of the challenges in the field and necessary next steps for deployment.

In the discussion: *Several limitations of our trial should be mentioned. First, our study was single center, reflecting the demographics and clinical practices of a particular population. Nevertheless, the AI model was trained on example images from another center and the clinical trial was performed as prospective external validation, suggesting*

generalizability of the AI techniques and workflow. Second, the study was not powered to assess long term outcomes based on differences in LVEF assessment. Although the results were consistent across subgroups, further analyses are needed to evaluate the impact of video selection, frame selection, and intra-provider variability. *Third, this trial used previously acquired echocardiogram studies, and while prospectively evaluated by sonographers and cardiologists, there can be bias when a different sonographer than the scanning sonographer interprets the images.* Finally, consistent with findings from most AI studies, we found model performance improvement scales with the number of training examples. Thus, we anticipate that future studies could improve upon the AI performance that we observed in the current study by implementing AI models developed based on an even greater number of training examples derived from a broad and diverse cohort of patients. *Notably, this clinical trial utilized an AI model entirely trained from an independent site, representing external validation of the model. Final deployment of AI models in cardiology will require additional regulatory oversight, clinician buy-in, and deep integration with clinical systems that need to be further studied.*

Reviewer Reports on the First Revision:

Referees' comments:

Referee #1 (Remarks to the Author):

The authors have done a better job of explaining their trial design and contextualizing it with actual prospective randomized trials, and their prior report of echo AI in cardiovascular medicine. Nonetheless, with the single center, blinded review of echocardiograms from 2019 supporting non-inferiority of ejection fraction and related sonographer interpretation and less than 2 minutes of saving time, it can hardly be asserted as a momentous advance or sign of readiness of use of deep neural networks instead of sonographers or cardiologists to interpret studies.

The statement "we think there should not be significant change in the results if the scanning was contemporaneous" can only be backed up by doing that work.

By using perviously acquired studies that are deemed evaluable, this does not mimic the real world that would be simulated by a prospective study. Despite the Cedars-Sinai IRB request to use previously obtained echocardiograms, there are many ways and reasons to proceed with a prospective trial that would not compromise patient care but provide more solid backing of the use of AI for this purpose.

Referee #2 (Remarks to the Author):

My technical comments have been properly addressed.

The main comment on the impact of the paper is indeed still open, and it is not clear if this can be addressed. I still believe that while the study is solid, its impact remains limited and it provides only a step towards the solution as discussed with the authors, while the full solution would be of much higher impact in this venue.

Referee #3 (Remarks to the Author):

The authors have satisfactorily addressed the points raised in the first round of reviews and further enhanced the manuscript. Even though the trial reported is limited to one specific AI model and one specific use case, the methods and results should pave way for future, more comprehensive trials of this nature, and as such represents an important early step towards eventual beneficial adoption of AI methods in clinical practice.

Author Rebuttals to First Revision:

Reviewer Comments

Referee #1 (Remarks to the Author):

The authors have done a better job of explaining their trial design and contextualizing it with actual prospective randomized trials, and their prior report of echo AI in cardiovascular medicine. Nonetheless, with the single center, blinded review of echocardiograms from 2019 supporting non-inferiority of ejection fraction and related sonographer interpretation and less than 2 minutes of saving time, it can hardly be asserted as a momentous advance or sign of readiness of use of deep neural networks instead of sonographers or cardiologists to interpret studies.

The statement "we think there should not be significant change in the results if the scanning was contemporaneous" can only be backed up by doing that work.

By using perviously acquired studies that are deemed evaluable, this does not mimic the real world that would be simulated by a prospective study. Despite the Cedars-Sinai IRB request to use previously obtained echocardiograms, there are many ways and reasons to proceed with a prospective trial that would not compromise patient care but provide more solid backing of the use of AI for this purpose.

Reply: Thank you. In this draft, we discuss the limitations and important caveats identified by the reviewer while also recognizing the difficulty of performing clinical trials in this space and the uniqueness of this trial in blinding and randomization. In the discussion:

Notwithstanding tremendous interest in AI technologies, there have been few prospective trials evaluating their efficacy and impact on clinician assessments. Important clinical trials of AI technology have already shown the efficaciousness of AI in cardiology^{21,25}, however given the difficulty of blinding a diagnostic tool, previous trials are often open label and compared with a placebo or no diagnostic assistance. Prior work have shown that there is can be Hawthorne effect when studying novel technologies like AI systems^{26,27+}. By introducing blinding with an active comparator arm, studies can better distinguish between the effect of the AI technology itself vs. the impact of being observed or the act of introducing an intervention.

...

In this trial, we utilized experienced sonographers as an active comparator vs. the AI for the initial assessment of LVEF, however different levels of experience and types of training can change the relative impact of AI compared to clinician judgement.

Referee #2 (Remarks to the Author):

My technical comments have been properly addressed.

The main comment on the impact of the paper is indeed still open, and it is not clear if this can be addressed. I still believe that while the study is solid, its impact remains limited and it provides only a step towards the solution as discussed with the authors, while the full solution would be of much higher impact in this venue.

Reply: Thank you. We agree that this trial is a stepping stone toward understanding AI in healthcare. Given the tremendous interest in medical AI and the limited number of randomized trials, the design and insights from this trial contribute to the critical question of how to evaluate medical AI. In addition to the particular results of the trial, generalizable and notable findings include clear evidence of anchoring bias for clinicians with diagnostic support, highlighting the importance of blinding and the need for active comparators for future AI trials, and contextualizing human clinician variation in the clinical workflow (largest study of human test-retest in echocardiography).

Notably, this clinical trial utilized an AI model entirely trained from an independent site, representing external validation of the model. Effective deployment of AI models in cardiology clinical practice will require additional regulatory oversight, adoption and appropriate use by clinicians, and functional integration with clinical systems – all of which need to be carefully considered and further studied.

Referee #3 (Remarks to the Author):

The authors have satisfactorily addressed the points raised in the first round of reviews and further enhanced the manuscript. Even though the trial reported is limited to one specific AI model and one specific use case, the methods and results should pave way for future, more comprehensive trials of this nature, and as such represents an important early step towards eventual beneficial adoption of AI methods in clinical practice.

Reply: Thank you so much for this feedback. We are excited that you liked the paper!